# Sustainable and Selective Extraction of Lipids and Bioactive Compounds from Microalgae

**DOI:** 10.3390/molecules24234347

**Published:** 2019-11-28

**Authors:** Ilaria Santoro, Monica Nardi, Cinzia Benincasa, Paola Costanzo, Girolamo Giordano, Antonio Procopio, Giovanni Sindona

**Affiliations:** 1Dipartimento di Ingegneria per l’Ambiente e il Territorio e Ingegneria Chimica, Università della Calabria, Cubo 45A, I-87036 Rende, Italy; girolamo.giordano@unical.it; 2Dipartimento di Scienze della Salute, Università Magna Græcia, Viale Europa, I-88100 Germaneto (CZ), Italy; pcostanzo@unicz.it (P.C.); procopio@unicz.it (A.P.); 3CREA Research Centre for Olive, Citrus and Tree Fruit, C.da Li Rocchi, I-87036 Rende, Italy; cinzia.benincasa@crea.gov.it; 4Dipartimento di Chimica e Tecnologie Chimiche, Università della Calabria, Cubo 12C, I-87036 Rende, Italy; giovanni.sindona@unical.it

**Keywords:** algal oil, green chemistry, green solvents, extraction, biofuel, bio compound

## Abstract

The procedures for the extraction and separation of lipids and nutraceutics from microalgae using classic solvents have been frequently used over the years. However, these production methods usually require expensive and toxic solvents. Based on our studies involving the use of eco-sustainable methodologies and alternative solvents, we selected ethanol (EtOH) and cyclopentyl methyl ether (CPME) for extracting bio-oil and lipids from algae. Different percentages of EtOH in CPME favor the production of an oil rich in saturated fatty acids (SFA), useful to biofuel production or rich in bioactive compounds. The proposed method for obtaining an extract rich in saturated or unsaturated fatty acids from dry algal biomass is disclosed as eco-friendly and allows a good extraction yield. The method is compared both in extracted oil percentage yield and in extracted fatty acids selectivity to extraction by supercritical carbon dioxide (SC-CO_2_).

## 1. Introduction

In recent years, the production of algae culture and usage of algal biomass conversion products have received much attention. Microalgae are a potential source of a wide range of high-value products for different biotechnological uses [1]. In particular, algae have long been considered excellent feedstock to produce oils. Algal oil, in fact, can be used in different sectors in addition to the production of biofuels [2], for example in the nutraceutical sector as nutritional supplements and in cosmetics.

The considerable amounts of lipid content in microalgae allow the production of alternative renewable cleaner fuels [3].

Biodiesel is a mixture of fatty acid alkyl esters usually obtained by transesterification (ester exchange reaction) of vegetable oils or animal fats [4]. Many research reports and articles have described different advantages of using microalgae for biodiesel production in comparison with other available feedstocks [5,6,7,8,9,10,11,12]. The lipid and fatty acid substances of microalgae differ in accordance with culture conditions. In fact, depending on the strain to which the algae belong, it can have between 20–80% of oil by weight of dry mass [13], and it also varies in the lipid composition [14]. 

From a practical point of view, microalgae are easy to cultivate, can grow with little or even no attention, use water unsuitable for human consumption, and are easily inclined to provide nutrients. 

The extracts of microalgae show antimicrobial, antiviral, antibacterial, and antifungal properties attributed to the presence of fatty acids [15] and are also used as ingredients in different skincare, sun protection, and hair care formulations. Microalgae are considered, in fact, as the predominant production sources for polyunsaturated fatty acids (PUFAs) that have to be supplied for the human diet [16,17]. PUFAs have been used in the prevention/treatment of cardiovascular diseases [18,19,20,21] and their derivatives, namely α-linolenic acid (ALA), eicosapentaenoic acid (EPA), docosapentaenoic acid (DPA), and docosahexaenoic acid (DHA), have also been reported for the treatment of type 2 diabetes, inflammatory bowel disorders, skin disorders, and asthma [22,23,24].

PUFAs play a major role in the treatment of arthritis, obesity, Parkinson’s disease, and heart disease [25]. EPA and DHA are the main derivatives of omega-3 fatty acids (PUFA n-3) and play a role in lowering blood cholesterol and in fetal brain development, respectively [26]. Carotenoids and pigments are the main constituents of microalgal-based food supplements, and they possess antioxidant activities with neuroprotective action and protection against chronic diseases [27].

Various extraction methods have been reported in the literature for micro-algal lipids. Conventional methods for extraction lipids include hexane extraction and vacuum distillation. The traditional solvent extraction is the most used method thanks to the simplicity in operation and the possibility of use in the industrial field [28]. Usually, solvents such as methanol and chloroform, and temperatures between 150 °C and 250 °C, are used to obtain high extractive yields of microalgae oil [29,30]. The Bligh and Dyer method is the one most used in the extraction and quantitation of lipids at the analytical level [29]. The use of flammable or toxic solvents is considered a very important problem due to the adverse health and environmental effects.

Over the years, several research groups have determined the profile of triglycerides in the oil extracted from microalgae, which is relevant for the production of biofuels [9,30,31,32,33]. New algae oil extraction techniques are being developed, such as enzyme-assisted extraction [34], microwave-assisted extraction [35], ultrasound-assisted extraction [36], pressurized liquid extraction [37], and supercritical fluid extraction [38,39]. Recent studies have shown that total lipids/bio-oil extraction from algae can occur through using supercritical carbon dioxide (SC-CO_2_) assisted with azeotropic co-solvents such as hexane and ethanol 1/1 at a reaction pressure of 340 bar, and a temperature of 80 °C in 60 min, obtaining a total algal lipid yield of 31.37% based on dry basis and a percentage of eicosapentanoic acid (EPA) in the range of 20% to 32% [40]. This procedure increased the total lipid yield and the selectivity, but the usage of hexane as a solvent lead to numerous consequences such as air pollution and toxicity.

Our research group has done extensive work on the identification and molecular characterization of food compounds [41,42,43]. We have developed environmentally friendly methods for the extraction and further chemical manipulation of natural bioactive molecules [44,45,46,47,48,49,50,51,52], reducing or eliminating the use and generation of harmful substances and solvents aiming to encourage green chemistry [53]. In this study, according to the previous studies based on the use of non-toxic solvents [54,55,56,57,58,59,60,61,62,63,64], we have focused our attention on the development of selective extractive processes for PUFAs rather than for saturated fatty acids and vice versa, using green solvents such as Cyclopentyl Methyl Ether (CPME) and ethanol (EtOH). 

CPME is an unconventional and an ethereal green solvent, which is very stable to peroxide formation, with low volatility and low water solubility [65]. Thus, it has increased interest as an industrial solvent [66]. It exerts no genotoxity or mutagenicity [66], and is produced by the addition of methanol to cyclopentene, with a 100% atom economy for its synthesis. It has also been studied in many important chemical processes including furfural synthesis [67] and extraction of natural products [68]. Previous works studied the liquid–liquid equilibria for ternary systems of water/CPME/alcohol (methanol, ethanol, 1-propanol, or 2-propanol) to test greener solvent systems that substitute and simulate the Bligh and Dyer method for oil extraction [69]. Recently, a comparative study of lipid extraction from wet microalgae was performed using several methods such as the Soxhlet, Bligh and Dyer, Folch, and Hara and Radin methods, with 2-methyltetrahydrofuran (2-MeTHF) and CPME as green solvents. The Bligh and Dyer methodology using the solvents 2-MeTHF/isoamyl alcohol (2:1 *v/v*) and CPME/methanol (1:1.7 *v/v*) showed an oil extraction yield less than 10% [70].

Thus, the objective of this work was to evaluate the use of a green binary solvent system CPME/EtOH for the selective extraction of lipids and bioactive compounds from microalgae dry. The various lipid components within each fraction have been characterized and quantified using GC-MS and LC-MS [71,72,73], following a standard procedure of transmethylation [74] available to our goal. The information on complete lipid characterization of extracts is essential for the successful selection of the extraction process that is useful for the production of biofuels or the development of potential nutraceuticals. 

## 2. Results and Discussion

The biodiesel or saturated fatty acids production process involves different steps, including lipid extraction and purification of fatty acids. The classic extraction processes often involve the use of toxic substances. Furthermore, the selective extraction processes that leads to obtaining oils rich in PUFAs or saturated fatty acids, and the separation of individual fatty acids, are difficult for the production of highly concentrated w-3 components. Docosahexaenoic acid (DHA, 22:6w3) is considered to be a crucial nutrient for fetal and infant development [75,76], and only recently researchers have developed the urea complexation to concentrate DHA from Crypthecodinium cohnii CCMP 316 biomass [77] and the production of algal oil enriched in w-3-PUFA [78,79] that is useful for human health. 

The use of SC-CO_2_ is an environmentally sustainable extraction method [39]. However, to have a higher extraction yield, it is necessary to use co-solvents such as hexane [40], invalidating the sustainability of the method itself. Moreover, this method is not always easily applicable at an industrial level. 

In the present work, according to the studies based on the use of non-toxic solvent [80,81], extracted lipids rich in PUFAs or saturated fatty acids by dried microalgae were obtained using CPME and ethanol as green solvents at different percentages. The optimization of the method was carried out by comparing the oil extraction yield obtained using different percentages of ethanol in CPME with the oil extraction yield obtained by SC-CO_2_ [38]. As can be shown in Figure 1, the extraction yield increased considerably with the use of 80% of ethanol in CPME (39.4%). 

The determination of fatty acid composition using GC-MS quantitative analysis in the extracted samples (see Appendix A) was performed. The GC-MS analysis of the methyl transesterified [82] algal oil was extracted using SC-CO_2_ (see Appendix A), revealing the presence of fourteen fatty acids (Table 1). At the time retention of 13.35 min, the peak of internal standard (methyl tricosanoate, 23:0) was observed.

Peak identification of the saturated fatty acids (SFA) and unsaturated fatty acids (PUFAs and MUFA) in the analyzed microalgae oil samples were carried out by comparison with retention time and mass spectra of known standards (Table 1). Samples were analyzed in triplicate.

From the GC-MS analysis, it was found that the most abundant fatty acid methyl esters (FAMEs) present in the various extracts were those of myristic and palmitic acid (two saturated fatty acids), and EPA and DHA (two ω-3 polyunsaturated fatty acids) derivatives. These were considered to evaluate the performance of the EtOH/CPME mixture compared to SC-CO_2_ in the extraction process (Figure 2).

It is evident that an oil richer in saturated fatty acids (Figure 2) has been obtained using a mixture of EtOH/CPME 8/2, the same solution solvent useful for obtaining a higher extraction yield of algal oil. 

On the contrary, an oil rich in EPA and DHA was obtained using a mixture of EtOH/CPME 6/4 (Figure 2), the same mixture of solvents, which showed a yield of the extracted oil equal to 32.8% against 30.0% yield obtained with SC-CO_2_. 

From the results of the quantitative analysis of all fatty acid methyl esters (see Appendix A), the yield percentage of all saturated fatty acids and all unsaturated fatty acids was calculated. 

Table 2 illustrates how the percentage of total SFA and total UFA (polyunsaturated and monounsaturated fatty acids) concentrations varied according to the percentage of EtOH in CPME (Table 2, entries 1–7) and using SC-CO_2_ (Table 2, entry 8). The use of SC-CO_2_ favored the isolation of UFA to SFA (66.93%). A comparable result was obtained using an EtOH/CPME 6/4 mixture (61.41%) useful for obtaining a higher oil extraction yield.

Electron ionization (EI) MS coupled to Gas chromatography (GC) to analyze fatty acids [83] has been usually applied but in addition to GC-MS, liquid chromatography (LC)-MS was also a useful method for the accurate analysis of profiling of FFAs [84,85,86] and for qualitative determination of nonvolatile compounds in food [87]. The use of electrospray ionization (ESI) MS, a soft ionization technique, provided the information on molecular ions. Therefore, tandem MS (MS/MS) was applied for the most sensitive and selective analysis of FFAs. 

To provide the identification of components in the extracted samples, liquid-chromatography mass spectrometry (LC-MS) was employed. ESI-MS/MS analysis was performed for all the ions present in the full scan chromatogram for each algae extract. 

The analytical technique thus confirmed in a more detailed way the presence of the previous extracted and previously quantified fatty acids. The chromatogram was obtained scanning between 50 and 800 amu in negative ion mode (Figure 3). 

From the information obtained in full scan mode, it was possible to have information about the molecular weights of the fatty acids possibly present in the samples under investigation.

From the molecular ions registered, the algal oil extract was composed of lauric acid (molecular ion at *m/z* 199.4); myristic acid (227.4); myristoleic acid (225.2); pentadecylic acid (241.4); palmitic acid (255.5); γ-linolenic acid (277.4); oleic acid (281.6); stearic acid (283.5); eicosapentanoic acid (301.6); docosaexaenoic acid (327.5); docosapentaenoic acid (329.7) and behenic acid (339.6). Successively, to attribute the molecular structures on each single molecular ion previously recorded, experiments in product ion scan (MS/MS) were performed. 

Table 3 lists the deprotonated molecules identified in full scan MS spectra, the fragment ions, and the precursor ions identified by MS/MS experiments. 

The initial complexity of the mass spectrum was, therefore, reduced when the product ion scan and a precursor ion scan were performed. ESI-MS/MS analysis was carried out for all the ions present in the full scan chromatogram for each algae extract. Ions included in the range of *m/z* 455–656 indicated molecular adducts between two identical or different fatty acid compounds (see Appendix A).

Based on this result, it was possible to choose the type of solvent/solvent mixture to be used depending on the type of extract to be obtained (richer than SFA or UFA). Both analytical techniques were useful and necessary to determine the type of fatty acids present in the extracted samples. Thus, it was possible to identify an environmentally sustainable extraction method for saturated or unsaturated fatty acids with excellent extraction yield and significant selectivity.

## 3. Materials and Methods 

### 3.1. Chemicals

Solvents, reagents, and thimbles were purchased from Sigma–Aldrich (Sigma–Aldrich, St. Louis, MO, USA). Dry algal biomass was provided by Aquafauna Biomarine inc. (P.O. Box 5, Hawthorne, CA, USA). 

### 3.2. Supercritical Fluid Extraction 

Fractionation of algal oil from algal biomass was carried out in a continuous extraction process supercritical fluid using an Applied Separations Speed SFE model 7070 apparatus (Applied Separations Inc., Allentown, PA, USA). About 5 g of sample and glass beads (size about 3 mm) were weighed and added to the vessel (1.27 cm i.d. × 25.4 cm long) and sealed with polypropylene wool at the top and the bottom of the extraction vessel. The oven temperature used was 80 °C and the vessel at a pressure of 35 MPa and room temperature of 25 °C. The extracted oils were collected within 10 mL vials. Each extraction was replicated 3 times.

### 3.3. Soxhlet Extraction

Soxhlet Extraction was carried out using 100 mL of solvent on 5 g of dried algae sample by Soxhlet apparatus. The extraction lasted for 1 h. The extract was filtered to remove possible solid particles. Organic solutions were then concentrated by rotary evaporation, and the traces of solvent in residual oil were removed by nitrogen flushing. The yield was calculated based on the weight of extracted oil and the weight of the start sample. The same process was repeated with different solvent mixtures; CPME 100%, CPME/EtOH (80:20, 60:40, 50:50, 40:60, 20:80) and EtOH 100%. Each extraction was replicated 3 times.

### 3.4. Preparation of Methyl Esters of Constituent Fatty Acids 

Fatty acids compositions were determined by their conversion to methyl esters. 15 mg of each oil was added to the internal standard (250 ng/100 µL chloroform, methyl tricosanoate, 23:0). The oil was subjected to transmethylation by treating 15 mg of each oil with 6 mL of 0.2 M sulfuric acid in methanol following standard procedure and 15 mg of hydroquinone [64]. The mixture was incubated for 12 h at 60 °C and subsequently cooled. 1 mL of distilled water was added to each vial and extracted 3 times with 1.5 mL of heptane. The organic phase containing the fatty acid methyl esters (FAMEs) was separated and evaporated under a stream of nitrogen. The FAMEs obtained were dried over anhydrous magnesium sulfate and kept for Gas Chromatography-Mass Spectrometry (GC-MS, Shimadzu, Kyoto, Japan) analysis.

### 3.5. GC-MS Analysis 

The FAMEs obtained from the different algal oils were analyzed on Shimadzu GC/MS-QP 2010 gas chromatography instrument with an autosampler AOC-20i (Shimadzu) equipped with a 30 m-QUADREX 007-5MS capillary column operating in the “split” mode, and 1 mL min-1 flow of He as carrier gas. The injector was operated at 250 °C, while the detector was operated at 380 °C. The oven temperature was programmed to rise from 70 to 135 °C at a heating rate of 2 °C/min, from 135 to 220 °C at a heating rate of 4 °C/min and from 220 to 270 °C at a heating rate of 3.5 °C/min. The mass spectrometer (Shimadzu, Kyoto, Japan) was operated in the electron impact (EI) mode at 70 eV in the scan range 50–500 m/z. The FAMEs were identified based on the authentic samples previously injected in combination with the examination with individual molecular weight, mass spectra, and comparison of fragmentation pattern in the mass spectrum with that of the National Institute of Science and Technology, NIST library.

### 3.6. Data Analysis 

Results were analyzed by using a one-way ANOVA (GraphPad Software Inc., San Diego, CA, USA), and data were presented as the mean ± standard error of mean (SEM), except otherwise indicated. All experiments were performed in triplicate. Values at *p* < 0.05 were taken as significant.

### 3.7. Flow Injection Analysis/Mass Spectrometry (FIA/MS)

The mass spectrometer system used for the qualitative analyses of the algal oil extracts was a Q-Trap API 4000 (MSD Sciex Applied BiosystemFoster City, CA). The analyses were performed by flow injection analyses (FIA) in both negative and positive ion modes under a flow rate of 10 µL/min. In general, the intensity of ions observed as positive ions were lower than those observed as negative ions. Therefore, the experiments were conducted using the negative mode. The most important point of this experiment was to confirm that ions generated from the target compounds were observed, rather than that merely any sort of ions were observed. Therefore, it was necessary to attribute observed ions to specific compounds. Fundamentally, in the attribution process, assuming that positive ions were protonated molecules [M + H]^+^ and negative ions were deprotonated molecules [M − H]^−^, it was verified whether they were consistent with the molecular mass of target compounds. The structural assignment was, therefore, based on the accurate mass of the pseudo-molecular ion [M − H]^−^, present in the negative ESI-MS chromatogram, and on the corresponding fragment ions detected by collision-induced dissociation (CID) under nitrogen (25% normalized collision energy) in the ion trap. The instrument parameters were set as follows: Ion spray voltage (IS) −4600 V; curtain gas 10 psi; ion source gas 12 psi; collision gas thickness medium; entrance potential 10 eV, declustering potential 70 eV, collision energy (CE) between 15 and 30 eV and collision exit potential (CXP) between 5 and 9 eV.

## 4. Conclusions 

In this work, a new eco-friendly and effective method for extracting total lipids/bio-oil from algae was shown. The maximum total algal lipid yield (39.4% based on dry basis) was obtained using a solution of EtOH/CPME (8:2) at a temperature of 80 °C and a reaction time of 60 min. This extraction condition is advantageous to obtain an oil rich in SFA, useful to biofuel production. An oily extract rich in bio-compounds that contribute to human well-being such as eicosapentaenoic acid, EPA (C20: 5, ω-3) and docosahexaenoic acid, DHA (C22: 6, ω-3) was obtained using a EtOH/CPME, 6:4, at the same temperature and reaction time. In these extraction conditions the oil extraction yield is equal to 32.8% against 30.0% if SC-CO_2_ is used.

This method offers many advantages over conventional extraction technologies, such as increased total lipid yield, increased selectivity, and preserved thermo-labile compounds. It is an alternative method for sustainable production of algal biofuel and the development of high-value co-products.

## Figures and Tables

**Figure 1 molecules-24-04347-f001:**
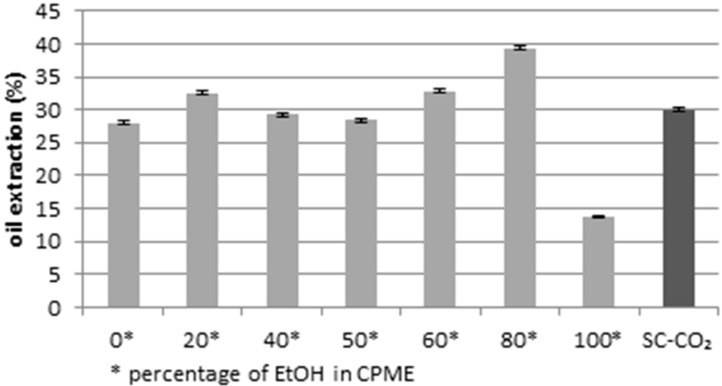
Oil extraction yields (g for 100 g of dry matter) using different volume percentages of ethanol in cyclopentyl methyl ether (CPME), compared to the yield obtained by SC-CO_2_. *p* < 0.05.

**Figure 2 molecules-24-04347-f002:**
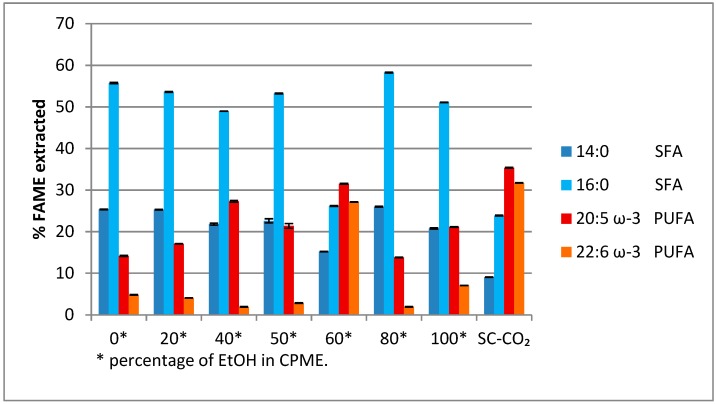
% of myristic acid, palmitic acid (saturated fatty acids, SFAs), eicosapentaenoic acid (EPA), and docosahexaenoic acid (DHA) (polyunsaturated fatty acids, PUFAs) extracted with a mixture ethanol (EtOH)/CPME and SC-CO_2_. *p* < 0.05.

**Figure 3 molecules-24-04347-f003:**
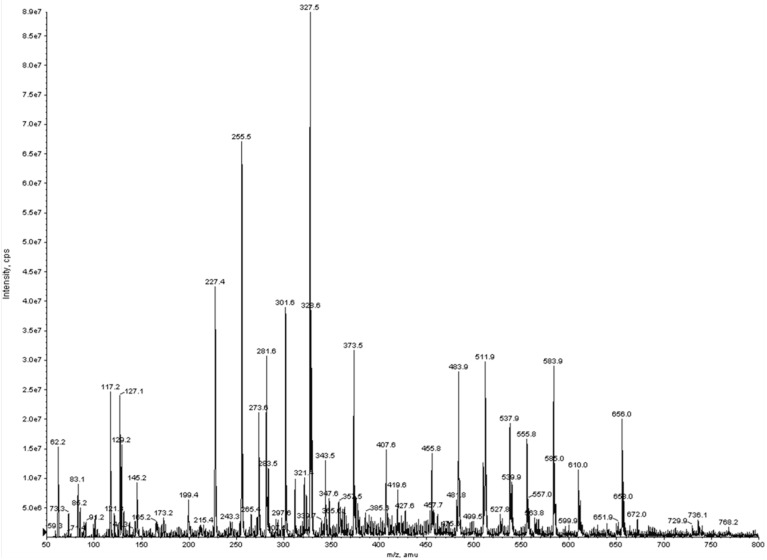
Negative electrospray ionization (ESI) full scan mass spectrum of algal oil sample.

**Table 1 molecules-24-04347-t001:** Fatty acids composition of algal oil by GC-MS analysis.

S/N	RT (min)	Name of Compound	Mol. Formula	Classification
1	1.5	Myristic acid (14:0)	C_14_H_28_O_2_	SFA
2	1.7	Pentadecylic acid (15:0)	C_15_H_30_O_2_	SFA
3	2.1	Palmitic acid (16:0)	C_16_H_32_O_2_	SFA
4	2.7	Heptadecanoic acid (17:0)	C_17_H_34_O_2_	SFA
5	3.1	Linoleic acid (18:2 ω-6)	C_18_H_32_O_2_	PUFA
6	3.2	γ-Linolenic acid (18:3 ω-6)	C_18_H_30_O_2_	PUFA
7	3.4	Oleic acid (18:1 ω-9)	C_18_H_34_O_2_	MUFA
8	3.5	Stearic acid (18:0)	C_18_H_36_O_2_	SFA
9	5.4	Eicosapentaenoic acid, EPA (20:5 ω-3)	C_20_H_30_O_2_	PUFA
10	5.5	Eicosatrienoic acid (20:3 ω-6)	C_20_H_34_O_2_	PUFA
11	5.6	Eicosatetraenoic acid (20:4 ω-6)	C_20_H_32_O_2_	PUFA
12	6.2	Eicosanoic acid (20:0)	C_20_H_40_O_2_	PSFA
13	9.1	Docosapentaenoic acid, DPA, (22:5 ω-3)	C_22_H_34_O_2_	PUFA
14	9.4	Docohexaenoic acid, DHA, (22:6 ω-3).	C_22_H_32_O_2_	PUFA

**Table 2 molecules-24-04347-t002:** SFA (saturated fatty acids) and UFA (unsaturated fatty acids) percentage concentration variation as a function of solvent/solvent mixture used.

Entry	Extraction Method	% SFA	% UFA
1	0 *	78.25 ± 0.16	21.75± 0.16
2	20 *	73.34 ± 0.04	26.66± 0.06
3	40 *	65.34 ± 0.09	34.66± 0.07
4	50 *	69.80 ± 0.15	30.20± 0.10
5	60 *	38.59 ± 0.03	61.41± 0.05
6	80 *	82.83 ± 0.13	17.17± 0.09
7	100 *	71.46 ± 0.01	28.54± 0.07
8	SC-CO₂ **	33.06 ± 0.18	66.93± 0.09

* Percentage of EtOH in CPME. Solvent mixture used for Soxhlet system. ** Supercritical carbon-dioxide extraction method. The values of percentages are in mean ± SD (*n* = 3).

**Table 3 molecules-24-04347-t003:** List of the deprotonated molecules identified in the full scan MS spectra of algal oil sample, fragment ions, and precursor ions identified in MS/MS spectra.

Analyte	[M − H]^−^	PIS	PREC
Lauric acid	199.4	181.6; 155.0	399.7; 455.9
Myristic acid	227.4	209.5; 183.3	455.7; 483.8; 509.8
Myristoleic acid	225.2	207.2; 181.5	482
Pentadecylic acid	241.4	223.2; 197.2	483.8
Palmitic acid	255.5	237.6; 211.6	511.8; 537.9; 583.9
γ-linolenic acid	277.4	259.4; 233.4	555.8
Oleic acid	281.6	263.4; 237.4	509.8; 537.9; 563.8
Stearic acid	283.5	265.4; 239.2	540.5; 568
Eicosapentanoic acid, EPA	301.6	283.6; 257.6	602.2
Docosahexaenoic acid, DHA	327.5	283.6; 309.2; 229.6	583.9; 610.0; 656.0; 658.0
Docosapentaenoic acid, DPA	329.7	311.7; 285.6	658
Behenic acid	339.6	321.3; 294.9	596.5; 569.2

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
