# Peer review of "Sustainable and Selective Extraction of Lipids and Bioactive Compounds from Microalgae"

_molecules, 2019, doi:10.3390/molecules24234347_

Round 1

Reviewer 1 Report

This article studies the extraction of lipids and bioactive compounds from microalgae using solvents of lower toxicity than the usual ones. The article is well written, and the research has been done appropriately. Some aspects, however, are suggested to improve the quality of the original manuscript:

1. Define the acronyms the first time they appear in the text, even when they are obvious. Do the same in the tables.

2. Figure 2 and Table 2 suggest the superiority of supercritical fluid extraction for the isolation of some compounds. This aspect should be also commented on the manuscript.

3. The results must be contextualized in the field, making a comparison table with other reported values.

4. In this sense, there is a recent article “Comparison of several methods for effective lipid extraction from wet microalgae using green solvents” https://doi.org/10.1016/j.renene.2019.04.168 that deserves to be considered.

5. The authors talk about LC/MS, but the chromatography (column, flow rate, mobile phase) is not described

Author Response

Review 1:

This article studies the extraction of lipids and bioactive compounds from microalgae using solvents of lower toxicity than the usual ones. The article is well written, and the research has been done appropriately. Some aspects, however, are suggested to improve the quality of the original manuscript:

Define the acronyms the first time they appear in the text, even when they are obvious. Do the same in the tables.

We defined the acronyms the first time they appear in the text and in the tables

Figure 2 and Table 2 suggest the superiority of supercritical fluid extraction for the isolation of some compounds. This aspect should be also commented on the manuscript.

In the text the following comment has been inserted to specify the superiority of the use of SC-CO2 to isolate PUFAs”The use of SC-CO2 favored the isolation of UFA to SFA (66,93 %). A comparable result was obtained using a EtOH /CPME 60/40 mixture (61,41 %) useful for obtaining a higher oil extraction yield”.

The results must be contextualized in the field, making a comparison table with other reported values. In this sense, there is a recent article “Comparison of several methods for effective lipid extraction from wet microalgae using green solvents” https://doi.org/10.1016/j.renene.2019.04.168 that deserves to be considered.

We reported the sentence “Recently, a comparative study of lipid extraction from wet microalgae was performed using several methods as the Soxhlet, Bligh and Dyer, Folch, and Hara and Radin methods, with  2-methyltetrahydrofuran (2-MeTHF) and CPME as green solvents. The Bligh and Dyer methodology using the solvents 2-MeTHF/isoamyl alcohol (2:1 v/v) and CPME/methanol (1:1.7 v/v) showed an oil extraction yield less than 10% [70]”. We added the reference 70: 70. De Jesus, S. S.; Ferreira, G. F.; Moreira, L. S.; Wolf Maciel, M. R.; Maciel Filho, R. Comparison of several methods for effective lipid extraction from wet microalgae using green solvents. Renew. Energy 2019, 143, 130–141.

The authors talk about LC/MS, but the chromatography (column, flow rate, mobile phase) is not described

The analyses were performed by flow injection analyses (FIA) in both negative and positive ion modes. We added the “under a flow rate of 10 μL/min” .

Reviewer 2 Report

In general, the manuscript describes an interesting research and it is well written. However, the experimental part must be completed with some more figures. Moreover, the discussion of the results should be improved in order to highlight the interesting findings of the work.  

Several minor corrections should be done through the manuscript, as shown below.

After these improvements, the present work can be published in Molecules.

Major revision:

- Some of the figures of the supplementary material should be shown in the text. Due to only 2 tables and 2 figures have been shown in the manuscript, some of the supplementary material should be added here to complete the information the reader see at a glance.

- Some chromatograms obtained with GC-MS and LC-MS for the PUFAs and SFA should be included in the manuscript for a better understanding.

- Lines 251-252: Check the results. In the figure 2 opposite results are found, in fact SC-CO2 extracts more than 60% with Soxhlet. Thus, the percentage obtained with Soxhlet should be lower or not? Please clarify this.

- The discussion of the results should be improved in order to highlight the interesting findings of the work.  

Minor revision:

- Line 21: Percentages

- Line 22: SFA should be defined.

- Line 22: Compounds bioactive should be replaced by bioactive compounds.

- Line 24: is disclosed is eco-friendly? English revision of the sentence is needed.

- Line 92: works

- Line 100: for developing

- Line 111: English revision of the sentence is needed.

- Figure 1: Remove border lines and horizontal lines behind the bars.

- Line 130: retention time

- Table 1: Highlight in bold or in a separate line the titles of the table (Name of compounds, Mol. Formula,…)

- Line 140: the performance of the mixture…

- Caption of figure 2: Write “saturated” instead of “satured”.

- Line 148: richer

- Table 2: Write “unsaturated” instead of “unsatured”.

- Table 2: Soxhlet

- Line 170: “the presence of the previous extracted and quanitified”.

- Line 185: an Applied

- Line 189: Remove the parenthesis after 25ºC.

- Line 190 and 198: times

- Line 192: algae

- Line 206: nitrogen. A colon is missing.

- Line 239: as follows

- Line 251: and reaction time

Author Response

Review 2:

In general, the manuscript describes an interesting research and it is well written. However, the experimental part must be completed with some more figures. Moreover, the discussion of the results should be improved in order to highlight the interesting findings of the work.  

Several minor corrections should be done through the manuscript, as shown below.

After these improvements, the present work can be published in Molecules.

 Major revision:

- Some of the figures of the supplementary material should be shown in the text. Due to only 2 tables and 2 figures have been shown in the manuscript, some of the supplementary material should be added here to complete the information the reader see at a glance. Some chromatograms obtained with GC-MS and LC-MS for the PUFAs and SFA should be included in the manuscript for a better understanding.

The figure 3 ( S1 in supplementary material; Negative ESI full scan mass spectrum of algal oil sample) and the Table 3 ( Table S2 in supplementary material; list of the deprotonated molecules identified in full scan MS spectra of algal oil sample, fragment ions and precursor ions identified in MS/MS spectra) have been reported in the manuscript..

- Lines 251-252: Check the results. In the figure 2 opposite results are found, in fact SC-CO2 extracts more than 60% with Soxhlet. Thus, the percentage obtained with Soxhlet should be lower or not? Please clarify this.

The yield of extracted oil is greater using Soxhlet. Otherwise Figure 2 indicates the selectivity of extracting compounds rather than others as a function of the solvent mixture used or the type of extraction. The SC-CO2 gives a lower oil extraction yield but the method favored the UFAs isolation to SFAs. This result is comparable to the result obtained when the EtOH / CPME 6/4 mixture with Soxhlet was used. Otherwise the EtOH / CPME 6/4 mixture not only gives me a higher oil extraction yield.

- The discussion of the results should be improved in order to highlight the interesting findings of the work.  

 We improved the discussion of the results.

Minor revision:

- Line 21: Percentages

we changed with “pententages”

- Line 22: SFA should be defined.

We defined SFA

- Line 22: Compounds bioactive should be replaced by bioactive compounds.

We replaced compounds bioactive with bioactive compounds.

- Line 24: is disclosed is eco-friendly? English revision of the sentence is needed.

The sentence has been modified. Is disclosed eco-friendly.

- Line 92: works

We changed works with works.

- Line 100: for developing

We changed for develop with for developing

- Line 111: English revision of the sentence is needed.

The sentence has been modified.

- Figure 1: Remove border lines and horizontal lines behind the bars.

The border lines removed.

- Line 130: retention time

We changed time retention with retention time.

- Table 1: Highlight in bold or in a separate line the titles of the table (Name of compounds, Mol. Formula,…)

In the table 1 the titles changed in bold in a separate line.

- Line 140: the performance of the mixture…

We changed the sentence “the performance the mixture” with “the performance of the mixture”.

- Caption of figure 2: Write “saturated” instead of “satured”.

We changed satured with saturated.

- Line 148: richer

We changed more rich with richer.

- Table 2: Write “unsaturated” instead of “unsatured”.

We changed unsatured with unsaturated.

- Table 2: Soxhlet

We changed shoxlet with soxhlet.

- Line 170: “the presence of the previous extracted and quanitified”.

The sentence has been modified.

- Line 185: an Applied

The sentence has been modified.

- Line 189: Remove the parenthesis after 25ºC.

The parenthesis after 25°C removed.

- Line 190 and 198: times

We changed time with times.

- Line 192: algae

We changed alga with algae

- Line 206: nitrogen. A colon is missing.

We added the colon.

- Line 239: as follows

We changed as follow with as follows.

- Line 251: and reaction time

We changed an reaction time with and reaction time.

Reviewer 3 Report

This work investigates the use of alternative green solvents for the extraction of lipids and bioactive compounds from microalgae. More specifically, different mixtures of ethanol and cyclopentyl methyl ether were evaluated and compared with supercritical fluid extraction using supercritical carbon dioxide. In this context, quantification by GC-MS and profiling by LC-MS, accompanied by MS/MS for identification, were employed to assess the effectiveness of this eco-friendly approach. The manuscript could be considered for publication after substantial revision, according to the following comments:

Line 18: Replace “have been used many times” with “have been frequently used over the years”.

Line 22: Change “SFA” to “saturated fatty acids (SFA)”. Also, change the phrase “…useful to production biofuel or rich of compounds bioactive” to “…useful to biofuel production or rich in bioactive compounds”.

Line 23: Replace the phrase “for obtain a rich extract of..” with the following: “for obtaining an extract rich in…”.

Line 24: Change “is disclosed is eco-friendly” to “is disclosed as eco-friendly”.

Line 36: Replace “lipids amount contents” with “amounts of lipids”. Change “allows” to “allow”.

Line 39: Insert “have” before “described”.

Line 43: Insert “in” before “the lipid composition”.

Line 49: Remove “a” before “predominant”.

Line 52: Do the authors refer to γ-linolenic acid (GLA) or α-linolenic acid (ALA)? Please correct using the appropriate abbreviation.

Line 59: Replace “to” with “with”.

Line 64: Replace “extractive” with “extraction”.

Line 67: Replace “for the…” with “due to the…”.

Lines 75-76: Change “a percentage eicosapentanoic acid” to “a percentage of eicosapentaenoic acid”. Replace “increase” with “increases”.

Line 77: Replace “lead” to “leads”.

Line 81: Replace “next” with “further”.

Lines 82-83: Change the phrase “…such as the Green Chemistry encouraging” as follows: “…aiming to encourage Green Chemistry”.

Line 84: Change “solvent” to “solvents”. Omit “a” before “selective extractive processes”

Line 85: Replace “rich in PUFA rather than…” with “for PUFAs rather than for…”. Change “solvent” to “solvents”.

Line 87: Replace “environmental” with “green”.

Line 88: Insert a comma after ”formation” and remove “and” before “with low volatility”. It is recommended to change “that” to “thus”.

Line 89: Replace “it is” with “it exerts”.

Line 92: Replace “workes” with “works”.

Line 94: Change “…to extraction of oil” to “…for oil extraction”.

Line 95: Replace “an environmental binary system” with “a green binary solvent system”.

Line 97: Change “has” to “have” and replace “quantificated” with “quantified”.

Line 98: Insert “a” before “standard procedure”.

Line 99: Remove “been”.

Line 100: Replace “process extraction” with “extraction process”. Insert “that is” before “useful” and “the” before “production”. Change “for develop…” to “the development…”.

Line 105: Change “PUFA” to “PUFAs”.

Line 107: Change “nutrients” to “nutrient”.

Line 109: Replace “enriched algal oil of…” with “algal oil enriched in…”.

Line 111: Replace “a most” with “an” and change “methods” to “method”.

Line 112: Change “seen” to “shown”.

Line 113: Replace “invaliding” with “invalidating”.

Line 115: Change “solvent” to “solvents”.

Line 116: Replace “lipid rich of PUFA” with “oils rich in PUFAs”.

Line 117: Change “percentage” to “percentages”.

Line 130: Change “At time retention…” to “At the retention time…”. Insert “the” before “peak”.

Line 131: Insert “is observed” after “(methyl tricosanoate, 23:0)”.

Line 133: Change “were” to “was”.

Line 134: Change “standard” to “standards”.

Line 137: Insert “it” before “was found”. Change “fatty acids methyl ester” to fatty acid methyl esters”.

Line 138: Remove “above all” and insert “those of” before “myristic…”.

Line 139: Change “derivative” to “derivatives”.

Line 140: Change “the performance the mixture EtOH/CPME” to “the performance of the EtOH/CPME mixtures”. Replace “comparing with” with “compared to”.

Lines 149-150: Place “extraction” before “yield of alga oil”.

Line 154: Insert “the” before “results”. Change “fatty acids methyl esters” to fatty acid methyl esters”.

Line 168: Replace “(LC/MS)” with “(LC-MS)”.

Line 171: Place “previously” before “quantified…”.

Line 174: Change “fatty acid” to “fatty acids”.

Line 176: Replace “extractive” with “extraction”.

Line 180: Add “were” before “purchased”.

Line 181: Add “was” before “provided”.

Line 184: Remove “for SC-CO2.

Line 185: Insert “extraction process” after “supercritical fluid…”. Replace “a” with “an” before “Applied…”.

Line 186: Change “was” to “were”.

Line 188: Change “is” to “was”. Please mention the pressure in the extraction vessel (in MPa).

Line 189: Change “with” to “within”.

Line 190: Replace “time” with “times”. Correct accordingly throughout the text (Lines 198

Line 195: Insert “the” before “weight”.

Line 202: Remove “the” before “each oil”

Lines 202-203: Replace “6 mL of methanolic solution 0.2 M of sulfuric acid” with “6 mL of 0.2 M sulfuric acid in methanol”.

Line 204: Replace “hour” with “h”.

Line 206: Insert a period after “nitrogen”.

Line 207: Change “was” to “were”.

Line 208: Replace “(GCMS)” with “(GC-MS)”.

Line 211: Insert a comma before “equipped”.

Lines 213, 216: Please indicate which were the actual injector and detector temperatures.

Line 215: Insert “and” before “from”.

Line 227: Change “analysis” to “analyses”.

Line 237: Change “fragments” to “fragment”.

Line 239: Change “follow” to “follows”.

Line 245: Replace “technique” with “method”.

Line 247: Insert “a” before “temperature”.

Line 248: Use “of” instead of “in”. Place “biofuel” before “production”.

Line 249: Replace “eicosapentanoic” with “eicosapentaenoic”.

Line 250: Use (C22:6, ω-3) for DHA instead of (C22: 5, ω-3). Remove “an” before “EtOH/CPME”.

Line 251: Replace “an” with “and”.

Line 254: Replace “a” with “an”.

Line 258: Replace “acids” with “acid”.

Table 1: Replace “Docohexanoic acid” with “Docohexaenoic acid”.

Figure 1: Please indicate any significant differences occurring among treatments based on ANOVA and comparisons of means at p < 0.05.

Figure 2. Replace “satured” with “saturated” in the figure caption. Also, insert “of” before “EtOH/CPME” (Lines 144-145). In addition, it would be useful to indicate any significant differences occurring among treatments based on ANOVA and comparisons of means (p<0.05).

References: Please consider using the abbreviated journal names throughout the section.

Author Response

Review 3:

This work investigates the use of alternative green solvents for the extraction of lipids and bioactive compounds from microalgae. More specifically, different mixtures of ethanol and cyclopentyl methyl ether were evaluated and compared with supercritical fluid extraction using supercritical carbon dioxide. In this context, quantification by GC-MS and profiling by LC-MS, accompanied by MS/MS for identification, were employed to assess the effectiveness of this eco-friendly approach. The manuscript could be considered for publication after substantial revision, according to the following comments:

Line 18: Replace “have been used many times” with “have been frequently used over the years”.

The sentence has been modified.

Line 22: Change “SFA” to “saturated fatty acids (SFA)”. Also, change the phrase “…useful to production biofuel or rich of compounds bioactive” to “…useful to biofuel production or rich in bioactive compounds”.

The sentence has been modified with “an oil rich in saturated fatty acids (SFA) useful to biofuel production or rich in bioactive compounds”

Line 23: Replace the phrase “for obtain a rich extract of..” with the following: “for obtaining an extract rich in…”.

The phrase has been replaced.

Line 24: Change “is disclosed is eco-friendly” to “is disclosed as eco-friendly”.

The sentence has been modified.

Line 36: Replace “lipids amount contents” with “amounts of lipids”. Change “allows” to “allow”.

We replaced and changed.

Line 39: Insert “have” before “described”.

We insert have before described.

Line 43: Insert “in” before “the lipid composition”.

We inserted in before the lipid composition.

Line 49: Remove “a” before “predominant”.

We removed a before “predominat”.

Line 52: Do the authors refer to γ-linolenic acid (GLA) or α-linolenic acid (ALA)? Please correct using the appropriate abbreviation.

We corrected-linolenic acid (ALA)

Line 59: Replace “to” with “with”.

We replaced “to” with “with”.

Line 64: Replace “extractive” with “extraction”.

We replace “extractive” with “extraction”

Line 67: Replace “for the…” with “due to the…”.

We replaced “for the” with “due to the”

Lines 75-76: Change “a percentage eicosapentanoic acid” to “a percentage of eicosapentaenoic acid”. Replace “increase” with “increases”.

The sentence changed and we replaced “increase” with “increases”

Line 77: Replace “lead” to “leads”.

We replaced "lead” with “leads”.

Line 81: Replace “next” with “further”.

We replaced “next” with “further”

Lines 82-83: Change the phrase “…such as the Green Chemistry encouraging” as follows: “…aiming to encourage Green Chemistry”.

The sentence changed.

Line 84: Change “solvent” to “solvents”. Omit “a” before “selective extractive processes”

We changed “solvent” with “solvents” and omitted “a” before “selective extractive processes”

Line 85: Replace “rich in PUFA rather than…” with “for PUFAs rather than for…”. Change “solvent” to “solvents”.

The sentence “rich in PUFA rather than…” replaced with “for PUFAs rather than for…”.  We changed “solvent” to “solvents”.

 Line 87: Replace “environmental” with “green”.

We replaced “environmental” with “green”.

Line 88: Insert a comma after ”formation” and remove “and” before “with low volatility”. It is recommended to change “that” to “thus”.

We inserted a comma after ”formation” and removed “and” before “with low volatility”. We changed “that” to “thus”.

Line 89: Replace “it is” with “it exerts”.

We replaced “it is” with “it exerts”.

Line 92: Replace “workes” with “works”.

We replaced “workes” with “works”.

Line 94: Change “…to extraction of oil” to “…for oil extraction”.

The sentence “…to extraction of oil” changed to “…for oil extraction”.

Line 95: Replace “an environmental binary system” with “a green binary solvent system”.

We replaced “an environmental binary system” with “a green binary solvent system”.

Line 97: Change “has” to “have” and replace “quantificated” with “quantified”.

We changed “has” to “have” and replaced “quantificated” with “quantified”.

Line 98: Insert “a” before “standard procedure”.

We inserted “a” before “standard procedure”.

Line 99: Remove “been”.

We removed “been”

Line 100: Replace “process extraction” with “extraction process”. Insert “that is” before “useful” and “the” before “production”. Change “for develop…” to “the development…”.

We replaced “process extraction” with “extraction process”, we inserted “that is” before “useful” and “the” before “production”. We changed “for develop…” to “the development…”.

Line 105: Change “PUFA” to “PUFAs”.

We replaced “PUFA” with “PUFAs”.

Line 107: Change “nutrients” to “nutrient”.

We replaced “nutrients” with “nutrient”.

Line 109: Replace “enriched algal oil of…” with “algal oil enriched in…”.

We replaced “enriched algal oil of…” with “algal oil enriched in…”.

Line 111: Replace “a most” with “an” and change “methods” to “method”.

We replaced “a most” with “an” and we changed “methods” to “method”

Line 112: Change “seen” to “shown”.

We replaced “seen” with “shown”

Line 113: Replace “invaliding” with “invalidating”.

We replaced “invaliding” with “invalidating”

Line 115: Change “solvent” to “solvents”.

We changed “solvent” to “solvents”.

Line 116: Replace “lipid rich of PUFA” with “oils rich in PUFAs”.

We changed lipid rich of PUFA” with “oils rich in PUFAs”.

Line 117: Change “percentage” to “percentages”.

We changed “percentage” to “percentages”.

Line 130: Change “At time retention…” to “At the retention time…”. Insert “the” before “peak”.

We changed “At time retention…” to “At the retention time…”. Insert “the” before “peak”.

Line 131: Insert “is observed” after “(methyl tricosanoate, 23:0)”.

We inserted “is observed” after “(methyl tricosanoate, 23:0)”.

Line 133: Change “were” to “was”.

We changed “were” to “was”.

Line 134: Change “standard” to “standards”.

We changed “standard” to “standards”.

Line 137: Insert “it” before “was found”. Change “fatty acids methyl ester” to fatty acid methyl esters”.

We inserted “it” before “was found” and we changed “fatty acids methyl ester” to fatty acid methyl esters.

Line 138: Remove “above all” and insert “those of” before “myristic…”.

We removed “above all” and inserted “those of” before “myristic…”.

Line 139: Change “derivative” to “derivatives”.

We changed “derivative” to “derivatives”.

Line 140: Change “the performance the mixture EtOH/CPME” to “the performance of the EtOH/CPME mixtures”. Replace “comparing with” with “compared to”.

We changed “the performance the mixture EtOH/CPME” to “the performance of the EtOH/CPME mixtures” and we replaced “comparing with” with “compared to”.

Lines 149-150: Place “extraction” before “yield of alga oil”.

We placed “extraction” before “yield of alga oil”.

Line 154: Insert “the” before “results”. Change “fatty acids methyl esters” to fatty acid methyl esters”.

We inserted “the” before “results” and we changed “fatty acids methyl esters” to fatty acid methyl esters”.

Line 168: Replace “(LC/MS)” with “(LC-MS)”.

We replaced “(LC/MS)” with “(LC-MS)”.

Line 171: Place “previously” before “quantified…”.

We placed “previously” before “quantified…”.

Line 174: Change “fatty acid” to “fatty acids”.

We changed “fatty acid” to “fatty acids”.

Line 176: Replace “extractive” with “extraction”.

We replace “extractive” with “extraction”

Line 180: Add “were” before “purchased”.

We added “were” before “purchased”.

Line 181: Add “was” before “provided”.

We added “was” before “provided”.

Line 184: Remove “for SC-CO2.

We removed “for SC-CO2”.

Line 185: Insert “extraction process” after “supercritical fluid…”. Replace “a” with “an” before “Applied…”.

We inserted “extraction process” after “supercritical fluid…”and we replaced “a” with “an” before “Applied…”.

Line 186: Change “was” to “were”.

We changed “was” to “were”.

Line 188: Change “is” to “was”. Please mention the pressure in the extraction vessel (in MPa).

We changed “is” to “was” and we mentioned the pressure in the extraction vessel (in MPa).

Line 189: Change “with” to “within”.

We changed  “with” to “within”.

Line 190: Replace “time” with “times”. Correct accordingly throughout the text (Lines 198)

We corrected time with times.

Line 195: Insert “the” before “weight”.

We inserted the before weight.

Line 202: Remove “the” before “each oil”

We removed  “the” before “each oil”.

Lines 202-203: Replace “6 mL of methanolic solution 0.2 M of sulfuric acid” with “6 mL of 0.2 M sulfuric acid in methanol”.

We replaced “6 mL of methanolic solution 0.2 M of sulfuric acid” with “6 mL of 0.2 M sulfuric acid in methanol”.

Line 204: Replace “hour” with “h”.

We replaced  “hour” with “h”.

Line 206: Insert a period after “nitrogen”.

We inserted a period after “nitrogen”

Line 207: Change “was” to “were”.

We changed  “was” to “were”.

Line 208: Replace “(GCMS)” with “(GC-MS)”.

We replaced “(GCMS)” with “(GC-MS)”.

Line 211: Insert a comma before “equipped”.

We inserted a comma before “equipped”.

Lines 213, 216: Please indicate which were the actual injector and detector temperatures.

We indicated the actual injector and detector temperatures.

Line 215: Insert “and” before “from”.

We inserted “and” before “from”.

Line 227: Change “analysis” to “analyses”.

We changed “analysis” to “analyses”.

Line 237: Change “fragments” to “fragment”.

We changed “fragments” to “fragment”.

Line 239: Change “follow” to “follows”.

We changed  “follow” to “follows”.

Line 245: Replace “technique” with “method”.

We replaced “technique” with “method”.

Line 247: Insert “a” before “temperature”.

We inserted “a” before “temperature”.

Line 248: Use “of” instead of “in”. Place “biofuel” before “production”.

We replaced “of” instead of “in” and “biofuel” before “production”.

Line 249: Replace “eicosapentanoic” with “eicosapentaenoic”.

We replaced “eicosapentanoic” with “eicosapentaenoic”.

Line 250: Use (C22:6, ω-3) for DHA instead of (C22: 5, ω-3). Remove “an” before “EtOH/CPME”.

We used (C22:6, ω-3) for DHA instead of (C22: 5, ω-3) and we removed “an” before “EtOH/CPME”.

Line 251: Replace “an” with “and”.

We replaced “an” with “and”.

Line 254: Replace “a” with “an”.

We replaced “a” with “an”.

Line 258: Replace “acids” with “acid”.

 We replaced “acids” with “acid”.

Table 1: Replace “Docohexanoic acid” with “Docohexaenoic acid”.

We replaced “Docohexanoic acid” with “Docohexaenoic acid”.

Figure 1: Please indicate any significant differences occurring among treatments based on ANOVA and comparisons of means at p < 0.05.

Figure 2. Replace “satured” with “saturated” in the figure caption. Also, insert “of” before “EtOH/CPME” (Lines 144-145). In addition, it would be useful to indicate any significant differences occurring among treatments based on ANOVA and comparisons of means (p<0.05).

We replaced “satured” with “saturated”.

Round 2

Reviewer 2 Report

Authors have followed every single suggestions of the reviewers. I think the manuscript can be published in present form in this journal.

Reviewer 3 Report

Thank you for the corrections introduced to the manuscript with your revision. I raise no more concerns about the paper.